# A leader-follower model for discrete competitive facility location problem under the partially proportional rule with a threshold

**Wuyang Yu** *

School of Management, Hangzhou Dianzi University, Zhejiang, China

* yu_wuyang@163.com

**Citation:** Yu W (2019) A leader-follower model for discrete competitive facility location problem under the partially proportional rule with a threshold. PLoS ONE 14(12): e0225693. https://doi.org/10.1371/journal.pone.0225693

**Data Availability Statement:** All relevant data are within the manuscript and its Supporting Information files.

**Funding:** The National Social Science Fund of China (17BGL182) supported this work.

## Abstract

When consumers are faced with the choice of competitive chain facilities that offer exclusive services, current rules do not properly describe the behavior pattern of these consumers. To eliminate the gap between the current rules and this kind of customers behavior pattern, the partially proportional rule with a threshold is proposed in this paper. A leader-follower model for discrete competitive facility location problem is established under the partially proportional rule with a threshold. Combining with the greedy strategy and the 2-opt strategy, a heuristical algorithm (GFA) is designed to solve the follower's problem. By embedding the algorithm (GFA), an improved ranking-based algorithm (IRGA) is proposed to solve the leader-follower model. Numerical tests show that the algorithm proposed in this paper can solve the leader-follower model for discrete competitive facility location problem effectively. The effects of different parameters on the market share captured by the leader firm and the follower firm are analyzed in detail using a quasi-real example. An interesting finding is that in some cases the leader firm does not have a first-mover advantage.

## 1 Introduction

In a market environment, competing facilities are those provided similar products or services for customers' patronage. In a market competition already exists or will exist in the future, the competitive facility location problem is to find the best locations for some of the new facilities. The goal of the competitive facility location problem is to maximize the market share captured by these facilities. Many competitive location models have been proposed for this purpose (see survey papers ([1–3]). These models can be categorized according to three kinds of features [4]: competition type, location space, and customer behavior.

Competition types are usually divided into two categories [5]: (1) static competition. This kind of competition is based on such assumption that the competitors already exist in the market, but they will not take action on new firms entering the market. (2) competition with foresight. There are two major situations in this kind of competition. One situation is when the firm wants to enter a virgin market in the knowledge that other competing actors will enter it soon. Another situation is when the firm wants to enter a competing market and considers the

**Competing interests:** The author has declared that no competing interests exist.

opponent's reaction (see [6]–[9]). Optimization models of the competitive facility location problems with foresight usually consist of the following two stages [5]: Both with the objective of maximizing market share, the leader locates new facilities in the first stage by considering the reaction of the follower, then the follower determines his facilities in the second stage. The location space is the second key ingredient, which is usually categorized as: (1) discrete space, where the location space is discrete and known in advance; (2) network space, assume that the customers and facilities are located in a given network; (3) plane space, where the new facilities can be located continuously on a two-dimensional plane. Most literature of the competitive facility location problem focuses on discrete space and less on network or plane space (see [10]–[15]).

Suppose there are several competitive facilities offer similar products or services, customer behavior refers to the way that a customer how to spend his buying power on these facilities. Hence, determining the type of customer behavior is the foundation to estimate the market share captured by a firm. Therefore, customer behavior plays a very important role in the location of competitive facilities. Different customer behaviors can be described in a precise manner with different rules. These rules are often expressed as functional forms of attraction that customers feel from different facilities [16]. Three customer behavior rules are usually employed in literature of the competitive facility location problem: (1) Binary Rule. This rule dates back to the duopoly model proposed by Hotelling [17]. It is assumed that the customer always patronizes the most attractive facility. (2) Proportional Rule. This rule is first addressed by Huff [18], which assumes that the customer patronizes each facility in proportion to its attraction. (3) Partially Binary Rule. Following the partially binary rule, the customer first selects the most attractive facility from each firm in the market, then splits his demand among those facilities proportionally to its attraction [19]. Some variants of these customer choice rules, mainly of the proportional rule, are proposed in some research works (see [20]–[28]). For example, Fernández et al. [27] proposed two new heuristic algorithms to solve competitive facility location problems with the binary rule and the partially binary rule. Qi et al. [28] considered a kind of customer behavior that customer only patronizes facilities within a range.

Ashtiani [29] indicated that some new features (such as relocation, design etc.) are included in the recent progress of competitive facility location problem. For example, Wang and Ouyang [30] presented a competitive service facility location design model by considering facility disruption risks. Nasiri et al. [31] studied the competitive facility location problem under the assumption of capacity constraints. Kung and Liao [32] constructed a discrete competitive facility location model considered the endogenous customer demands and network effects. Casas-Ramírez et al. [33] studied the competitive facility location problem with the customers' patronize behavior based on a predetermined list of preferences. Zhang et al. [34] studied the competitive facility location problem in the Stackelberg game framework. In their paper, facilities face disruption risks, and each customer patronizes the nearest operational facility.

Table 1 lists some of the relative studies, classifying them in terms of competitive type (Foresight (F) and Static (S)), location space (Discrete (D), Network (N), and Plane (P)), and customer behavior (Binary (B), Proportional (P), and Partially binary (PB)).

The above three rules can describe most of the customer behavior in the competitive facility location problem. However, when customers faced with the choice of competitive chain facilities offer exclusive services, none of these three rules can describe the customer behavior involved very well. For example, when somebody wants to apply for a bank account for deposit/withdrew or other services, usually he (or she) compares different banks at first then selects the most attractive bank to obtain the bank card. Because each bank has several business service points (including manual service points and ATMs), customers usually use the total

**Table 1. Selected researchers and classification.**

| Authors | Competitive | | Location | | | Behavior | | |
|---|---|---|---|---|---|---|---|---|
| | (F) | (S) | (D) | (N) | (P) | (B) | (P) | (PB) |
| Ashtiani et al. [5] | ✓ | | ✓ | | | | ✓ | |
| Beresnev [23] | ✓ | | ✓ | | | ✓ | | |
| Biesinger et al. [26] | | ✓ | ✓ | | | ✓ | ✓ | ✓ |
| Casas-Ramírez et al. [33] | ✓ | | ✓ | | | ✓ | | |
| Drezner et al. [7] | | ✓ | ✓ | | | | ✓ | |
| Fernández J. et al. [25] | ✓ | | | | ✓ | | | ✓ |
| Fernández J. et al. [14] | | ✓ | | | ✓ | | | ✓ |
| Fernández J. et al. [15] | | ✓ | | | ✓ | | ✓ | |
| Fernández P. et al. [27] | | ✓ | ✓ | | | ✓ | | ✓ |
| Grohmann et al. [11] | | ✓ | | ✓ | | | ✓ | |
| Kung et al. [32] | | ✓ | ✓ | | | | ✓ | |
| Mirzaei et al. [36] | ✓ | | ✓ | | | ✓ | | |
| Nasiri et al. [31] | ✓ | | ✓ | | | ✓ | | |
| Qi et al. [28] | ✓ | | ✓ | | | | ✓ | |
| Shiode et al. [12] | ✓ | | | ✓ | | | ✓ | |
| Shiode et al. [13] | ✓ | | | ✓ | | | ✓ | |
| Zhang et al. [34] | ✓ | | ✓ | | | ✓ | | |

attraction of all business service points to evaluate the bank. After getting the bank card, he (or she) will patronize each business service points of this bank in proportion to its attraction. The process of a customer applies for a bank account can be regarded as a typical description of customer behaviors when customers facing competitive facilities that offer exclusive services. This kind of customer behavior pattern can also be observed in another scene. Suppose the consumer plans to do a lot of activities such as shopping, entertainment, eating, etc. at a week-end, he (or she) is likely to choose a comprehensive mall where all these activities can be performed. If there are multiple optional comprehensive malls, he (or she) will compare the total appeal of these malls and then chooses the one that is most attractive to him (or her) to carry out the planned activities. When we consider the comprehensive mall as a firm and treat the different services as different facilities, we will find that the consumer choice behavior pattern is similar to the previous example. Unfortunately, none of the binary rule, the proportional rule, and the partially binary rule can illustrate this kind of customer behavior very well. To eliminate the gap between current rules and this kind of customer behavior is the motivation of our paper. In fact, the partially proportional rule with a threshold is proposed for this purpose.

The remainder of the paper is organized as follows: Section 2 is devoted to present the definition of the partially proportional rule with a threshold and to propose a competitive facility location model with foresight. Section 3 proposes an improved ranking-based greedy algorithm to solve the presented model. Section 4 illustrates the effectiveness of the algorithm through numerical tests, the effects of different factors on the market share are also analyzed in a quasi-real example. Finally, some conclusions are presented in section 5.

## 2 Model description

### 2.1 Customer choice behavior

In this section, we first propose a suitable rule to describe the behavior of customers when they faced the choice of competitive chain facilities that offer exclusive services. Suppose there are

several firms in the competitive market, each firm offers exclusive service through its facilities. The attraction of a firm is defined as the sum of attraction of its facilities. A customer only chooses one of these services due to the exclusiveness of these services. If a customer chooses one of the firms, he (or she) can be serviced by any chain facility of this firm. In order to describe well the behavior pattern of customers when faced with this choice. The **Partially Proportional Rule** defined as follows: A customer chooses the most attractive firm from all firms at first, then splits his (or her) demands on facilities of the selected firm in proportion to its attraction. In addition, in many cases, when the attractiveness of two firms is not much different for the customer, the customer will think that the two firms have o essential difference, and therefore any firm will be selected for consumption with the same probability. Hence, the **Partially Proportional Rule with a Threshold** can be regarded as a generalized version of the former defined rule. That is, let the maximal attraction of all firms be $G$. For firms with attraction in the interval $[G - \delta, G]$, the customer selects any of these firms with the same probability and then patronizes the facilities of the selected firm in proportion to their attractiveness.

We present an example to illustrate the partially proportional rule with a threshold. Suppose that there are three firms A, B, and C in the competing market. Firm A has four facilities 1, 4, 5, 9; firm B has three facilities 3, 6, 7; and firm C has three facilities 2, 8, 10. Suppose that for the customer, the attraction $a(k)$ of each facility $k$ is equal to its number, i.e. $a(k) = k$. The patronizing patterns under the partially proportional rule with different thresholds are shown in Fig 1 as follows.

In Fig 1, the total attraction of the firm A, B, C are 19, 16 and 20, respectively. Hence, if the customer following the partially proportional rule with a threshold $\delta = 0.5$, he chooses the firm C and splits his demand on facilities 2, 8, 10 in proportional to 2/20, 8/20 and 10/20. If the customer following the partially proportional rule with the threshold $\delta = 1$, then firm A and C are no different for this customer. Hence, the customer first selects firm A or C with the same probability, i.e. $p = q = 1/2$, and then patronizes the facilities of the selected firm in proportion to its attractiveness.

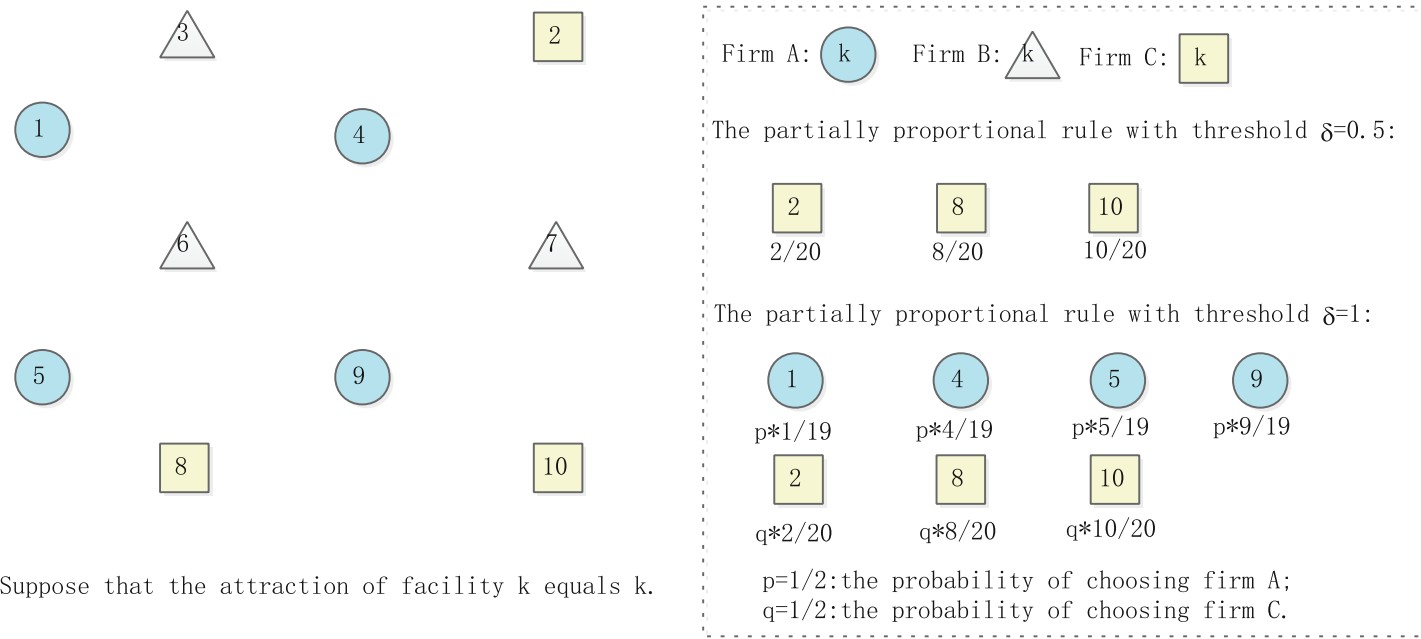

**Fig 1. Example for the partially proportional rule with a threshold.**

## 2.2 Proposed model

A two-dimensional market region is considered in this paper. Suppose that the demand is concentrated on $M$ points. Two competitive firms are referred to as the leader and the follower. The leader intends to open $p$ chain facilities in the set of potential facility locations, with belief that the follower will respond to his action by launching $q$ facilities. It is assumed that only one facility can be opened at each potential location. Both firms offer exclusive services through their chain facilities to compete for market share. So the partially proportional rule with a threshold is suitable to describe the customer behavior for this kind of competitive facility location problem.

The following notations are used:

$i$, $I$: index and set of demand points (customers);

$J_L$: potential facility locations of the leader firm;

$J_F$: potential facility locations of the follower firm;

$\delta$: the threshold in the partially proportional rule;

$w_i$: demand of the customer $i$;

$d_{ij}$: distance between customer $i$ and facility $j$;

$q_{ij}$: quality that customer $i$ feels from facility $j$;

$p$: the number of new facilities that the leader firm want to open;

$q$: the number of new facilities that the follower firm want to open;

$y_j^L, j \in J_L$: binary variable $y_j^L = 1$ if new facility $j$ is open by leader firm and 0 otherwise;

$y_j^F, j \in J_F$: binary variable $y_j^F = 1$ if new facility $j$ is open by follower firm and 0 otherwise;

$M_i^L$: the proportion of demand that the leader firm captured from customer $i$;

$M_i^F$: the proportion of demand that the follower firm captured from customer $i$.

In this paper, the facility attractiveness level is relate to quality and a reverse one with the distance between the facility and the customer. According to this, the attractiveness level of facility $j$ for customer $i$ equals:

$$A_{ij} = \frac{q_{ij}}{1 + (d_{ij})^{\alpha}} \tag{1}$$

In Eq (1), 1 is added to $(d_{ij})^{\alpha}$ to avoid the denominator becoming 0 when the distance between the facility and the customer is 0. The parameter $\alpha$ usually takes the value 1 or 2. The total attractions that customer $i$ feels from the leader firm and the follower firm are $\sum_{j \in J_L} A_{ij} y_j^L$ and $\sum_{j \in J_F} A_{ij} y_j^F$, respectively. Suppose that the location plan $(y^L)$ of the leader firm has been determined, then the proportion of demand that the follower firm captured from customer $i$ can be expressed as:

$$M_i^F(y^L, y^F) = \begin{cases} 1, & \sum_{j \in J_F} A_{ij} y_j^F > \sum_{j \in J_L} A_{ij} y_j^L + \delta \\ \frac{1}{2}, & |\sum_{j \in J_F} A_{ij} y_j^F - \sum_{j \in J_L} A_{ij} y_j^L| \le \delta \\ 0, & \sum_{j \in J_F} A_{ij} y_j^F < \sum_{j \in J_L} A_{ij} y_j^L - \delta \end{cases} \tag{2}$$

Obviously, the proportion of demand that the leader firm captured from customer $i$ is $M_i^L = 1 - M_i^F$. Therefore, the competitive facility location problem with foresight can be formulated as the following bi-level programming model:

**Upper level (Leader Model):**

$$Z^L = \max \sum_{i \in I} w_i M_i^L(y^L, y^F) \tag{3}$$

$$s.t. \begin{cases} \sum_{j \in J_L} y_j^L = p \\ y_j^L \in \{0, 1\}, j \in J_L. \end{cases} \tag{4}$$

Here, $y^F$, for each value of $y^L$, is the optimal solution of the lower level problem:

**Lower level (Follower model):**

$$Z^F(y^L) = \max \sum_{i \in I} w_i M_i^F(y^L, y^F) \tag{5}$$

$$s.t. \begin{cases} \sum_{j \in J_F} y_j^F = q \\ y_j^F + y_j^L \le 1, j \in J_F \cap J_L \\ y_j^F \in \{0, 1\}, j \in J_F. \end{cases} \tag{6}$$

Although the expression of $M_i$ looks difficult to establish linear model, the follower problem can be formulated as an integer model by introducing two 0-1 variables $x^+$ and $x^-$:

$$Z^F(y^L) = \max \sum_{i \in I} w_i \frac{x_i^+ + x_i^-}{2} \tag{7}$$

$$s.t. \begin{cases} \sum_{j \in J_F} y_j^F = q \\ x_i^+ \left( \sum_{j \in J_L} A_{ij} y_j^L + \delta \right) < \sum_{j \in J_F} A_{ij} y_j^F, i \in I \\ x_i^- \left( \sum_{j \in J_L} A_{ij} y_j^L - \delta \right) \le \sum_{j \in J_F} A_{ij} y_j^F, i \in I \\ y_j^F + y_j^L \le 1, j \in J_F \cap J_L \\ x_i^+, x_i^-, y_j^F \in \{0, 1\}, i \in I, j \in J_F. \end{cases} \tag{8}$$

Note that if $\sum_{j \in J_L} A_{ij} y_j^L + \delta < \sum_{j \in J_F} A_{ij} y_j^F$ then maximizing the objective function makes $x_i^+ = 1$ while at the same time $\sum_{j \in J_L} A_{ij} y_j^L - \delta \le \sum_{j \in J_F} A_{ij} y_j^F$ implies $x_i^- = 1$. In this case, the follower firm get the total demand of customer $i$. If $\sum_{j \in J_L} A_{ij} y_j^L - \delta \le \sum_{j \in J_F} A_{ij} y_j^F \le \sum_{j \in J_L} A_{ij} y_j^L + \delta$, then $x_i^+ = 0$ and $x_i^- = 1$ are true, the proportion of demand that the follower firm captured from customer

$i$ is 1/2. At last, $\sum\limits_{j \in J_F} A_{ij} y_j^F < \sum\limits_{j \in J_L} A_{ij} y_j^L - \delta$ implies that $x_i^+ = 0, x_i^- = 0$, the demand of customer $i$ is totally captured by the leader firm.

## 3 Solution method

Hansen et al. [35] indicated that the bi-level programming model is NP-hard even for the simplest linear-linear problem. Bi-level programming problems are very sensitive to constraints, so the solution space is completely changed by the very different constraints in our lower-level model (7-8). Mirzaei et al. [36] introduced four types of exact algorithms to solve the bi-level competitive facility location problem. But these algorithms cannot solve our problem directly due to the special structure of our lower-level model. In addition, for large scale problems, these algorithms cannot get an exact solution in a short time.

Two heuristic algorithms that based on ranking strategy are proposed by Fernández et al. [27]. Numerical tests show that the ranking strategy is effective to find near optimal solutions of the static competitive facility location problems. In order to solve the leader-follower model of the competitive facility location problem under the partially proportional rule with a threshold. We adopt the ranking-based heuristic algorithm as the framework of the main algorithm. Then for any given solution of the leader model, an efficient sub-algorithm is presented to solve the follower model. This sub-algorithm is designed on the greedy strategy and the 2-opt strategy as follows:

**Greedy-based Follower Algorithm (GFA)**:

```
0. Input a solution of the leader firm y^L and the lower bound Z̲^F;
1. Let y_j^N := 0,∀j ∈ J_F, and let J_F^1 = {j ∈ J_F|y_j^L = 1}; % here y_j^N is used to
indicate the current location of the follower firm;
2. While q ≥ 1 do:
      Let J_F^N := {j ∈ J_F|y_j^N = 0}\J_F^1; % here J_F^N is used to represent the set of
all current optional locations of the follower firm;
      For each k ∈ J_F^N:
         Let y_k^F := 1,y_j^F := 0,∀j ∈ J_F,j ≠ k;
         Let y^F := y^F + y^N calculate Z_k^F ≜ Z^F(y^L,y^F);
         If Z_k^F ≥ Z̄^F,
            then stop and return ỹ^F := y^F, Z̃^F := Z^F(y^L,ỹ^F);
         End if;
      End for;
      Select ỹ^F := argmax{Z_k^F|k ∈ J_F^N}, and let Z̃^F := Z^F(y^L,ỹ^F),y^N := ỹ^F and
q := q - 1;
3. End while;
4. Let D_1 := {j ∈ J_F|ỹ^F = 1},D_0 := {j ∈ J_F|ỹ^F = 0}\J_F^1;
5. For each i ∈ D_1
      For each j ∈ D_0
         y_temp := ỹ^F,y_temp(i) := 0,y_temp(j) := 1; Calculate z_temp := Z_j^F(y^L,y_temp);
         If Z̃^F < z_temp,
            then ỹ^F := y_temp,Z̃^F := z_temp;
         End if;
      End for;
   End for;
6. Stop and return ỹ^F and Z̃^F.
```

Note that (GFA) is an algorithm based on the greedy strategy and the 2-opt strategy. The step (2-3) is the greedy strategy that follower chooses the best facility one by one. The step 5 of the algorithm (GFA) is the 2-opt strategy. Note that if we take $\bar{Z}^F := \infty$, it is obvious that the

result of the algorithm (GFA) is not affected by this lower bound. In fact, the underlined parts about the lower bound $\bar{Z}^F$ are used to accelerate the algorithm from the viewpoint of the leader firm. The lower level problem can be solved quickly by this algorithm while maintaining a relatively high approximation of the solution. The comparison of the algorithm (GFA) and the direct method to solve model (7-8) is presented in the next section.

Before we give the algorithm to solve the problem, let's take a look at the following conclusion: Suppose that among all the leader's facility location schemes currently known, the maximal market share captured by the leader firm is $\bar{Z}^L$. Then for any new solution of the leader ($\tilde{y}^L$), once there is a solution of the follower ($y^F$) such that $\sum_{i \in I} w_i M_i^L(\tilde{y}^L, y^F) = \sum_{i \in I} w_i(1 - M_i^F(\tilde{y}^L, y^F)) < \bar{Z}^L$, the optimal solution of the leader cannot be $\tilde{y}^L$, this is equivalent to the abort condition in step 2 of the sub-algorithm (GFA): $Z_k^F \geq \bar{Z}^F$. Based on the above conclusion, we proposed an improved ranking-based greedy algorithm by embedding the algorithm (GFA) as follows:

**Improved Ranking-Based Greedy Algorithm (IRGA):**

```
1. Input the parameter T, which indicates the number of iterations of
   the algorithm. Generate a random solution yᴸ ∈ D(L), call algorithm
   (GFA) to get ỹ and Z̃ᶠ by inputting the leader's solution yᴸ and the
   lower bound Z̄ᶠ := ∞, then update Z̄ᶠ := Z̃ᶠ(yᴸ, ỹᶠ);
2. While T > 0, do:
     Let  Y¹ = {j ∈ Jₗ|yⱼᴸ = 1}, Y⁰ = {j ∈ Jₗ|yⱼᴸ = 0};
     Randomly choose j₁ ∈ Y¹, j₂ ∈ Y⁰, let ỹᴸ := yᴸ, ỹᴸ(j₁) := 0, ỹᴸ(j₂) := 1;
     Call (GFA) with ỹᴸ and the lower bound Z̄ᶠ to get ỹᶠ and Z̃ᶠ(ỹᴸ, ỹᶠ),
   update Z̄ᶠ := min{Z̄ᶠ, Z̃ᶠ(ỹᴸ, ỹᶠ)}.
     If Zᶠ(ỹᴸ) < Zᶠ(yᴸ),
        then rⱼ₁ := rⱼ₁ - 1, rⱼ₂ := rⱼ₂ + 1;
        else rⱼ₁ := rⱼ₁ + 1, rⱼ₂ := rⱼ₂ - 1;
     End if;
     If any rⱼ = 0,
        then set rⱼ := rⱼ + 1 for all j ∈ Jₗ;
     End if;
     Generate a new solution yᴸ ∈ D(L) according to the probability
   pⱼ := rⱼ/∑_{j∈Jₗ} rⱼ, and update T := T - 1.
3. End while.
4. Return yᴸ, Zᶠ(yᴸ), and Zᴸ = ∑_{i∈I} wᵢ - Zᶠ(yᴸ).
```

From (IRGA), we know that the lower bound $\bar{Z}^F$ interacts between the leader problem and the follower problem. By updating the minimal known $\bar{Z}^F$ timely, numerous calculating process can be eliminated while maintaining the optimality of the solution.

## 4 Numerical example

In this section, we first provide computational experiments to evaluate the performance of the proposed algorithm. For this purpose, we generate some problems with different scales to test the proposed algorithm. After proving the effectiveness of the algorithm, we present a quasi-real example to analysis affections of different factors.

### 4.1 Performance of the algorithm

To evaluate the performance of the algorithm (IRGA), we make three type of numerical tests as follows: (1) Test the performance of the sub-algorithm (GFA), the key performance measure

is the average gap between $Z^F(y^L)$ given by (GFA) and $Z^F_{exact}(y^L)$ given by the exact method, where $y^L$ is given randomly; (2) Test the performance of the algorithm (IRGA) for small scale problems, the key performance measure is the average gap between the value of $Z^L$ obtained by (IRGA) and the exact optimum $Z^L_{exact}$ obtained by the enumeration method; (3) Test the performance of the algorithm (IRGA) for large scale problems, the key performance measure is the gap between the average value of $Z^L$ and the maximum value of $Z^L$ for multiple random calculations.

For the first type of tests, we generate 9 problems with different scales to test the effectiveness of (GFA). In these random test problems, customers and potential facility locations are all locate in the region $[1, 100] \times [1, 100]$. The coordinates of these points are taken from the integer points of this region in random. The qualities of the facilities are taken as 1. The demands of customers' $w_i$ are random integer numbers chosen from the interval $[0, 100]$. The distance between two points $(x_i, y_i)$ and $(x_j, y_j)$ is the Euclidean distance $(d_{ij} = \sqrt{(x_i - x_j)^2 + (y_i - y_j)^2})$. The distances used in numerical tests are relatively small due to the limitation of the region, so we set the value of $\alpha$ to 2 to avoid the situation where the attractiveness of firms are too close. According to subsection 2.2, we know that the equivalent model (7-8) of the follower problem can be solved directly using some commercial software. For each test problem, we generate 10 solutions for the leader's firm in random, then compare solutions of the follower's problem that get from the direct method and the algorithm (GFA). The direct method is implemented by using the Cplex software. The algorithm (GFA) is implemented in the Matlab R2015b and run on a portable computer with an Intel Core i5 Processor (2.5 GHZ) and 8 GB memory. Comparisons of this direct method and the algorithm (GFA) are presented in Table 2. Note that here the lower bound is $\bar{Z}^F := \infty$.

In Table 2, the average times of the direct method and the algorithm (GFA) are $T_D$ and $T_G$, respectively. For a given leader's solution $y^L$, suppose that the market shares captured by the follower according to the direct method and the (GFA) are $Z^F_D(y^L)$ and $Z^F_G(y^L)$, respectively. For these randomly generated leader's solutions $y^L$, the $Gap_1$ in Table 1 is the average of $gap(y^L)$ which is defined as follows: $gap(y^L) := \dfrac{Z^F_D(y^L) - Z^F_G(y^L)}{Z^F_D(y^L)}$. From Table 2, we know that solutions get from the algorithm (GFA) are very close to the corresponding exact solutions get from the direct method. On the other hand, the times required by the algorithm (GFA) is dramatically less than the times required by the direct method, especially when the scales of problems are relatively large. It can also be seen from Table 2 that for a given solution of the leader firm, the average gap between the sub-algorithm (GFA) and the exact method increases with $q$. Alekseeva et al. [37] observed that under fixed value of $M$ and $N$, the leader-follower competitive facility location problem becomes more difficult when $p = q = [N/3]$. In this case, a lot of calculation time is required to solve the follower's problem and check the feasibility of the bi-level structure. Most of the literature mentioned in Table 1 that considers foresight can only solve small scale problems within $q < 10$. So from a perspective of computation time, we

**Table 2. Comparison of direct method with greedy-based follower algorithm ($p = q$).**

| | $q = 5$ | | | $q = 10$ | | | $q = 15$ | | |
|---|---|---|---|---|---|---|---|---|---|
| $(M, N)$ | (100,30) | (150,30) | (200,30) | (100,40) | (150,40) | (200,40) | (100,50) | (150,50) | (200,50) |
| $Gap_1$ | 0 | 0.16% | 0.16% | 0.16% | 0.14% | 0.15% | 0.74% | 1.13% | 0.42% |
| $T_D(s)$ | 2.38 | 10.60 | 14.79 | 20.46 | 168.34 | 289.07 | 77.15 | 1718.01 | 3632.15 |
| $T_G(s)$ | 0.0296 | 0.0621 | 0.0390 | 0.1318 | 0.1380 | 0.1611 | 0.2757 | 0.3179 | 0.3225 |

**Table 3. Optimal performance tests of the (IRGA) ($M$ = 50, $N$ = 20).**

| $(p, q)$ | (3, 3) | (3, 4) | (3, 5) | (4, 3) | (4, 4) | (4, 5) | (5, 3) | (5, 4) | (5, 5) | (6,3) | (6,4) | (6,5) |
|---|---|---|---|---|---|---|---|---|---|---|---|---|
| $Z^L_{max}$ | 1374 | 997 | 873 | 1642 | 1414 | 952 | 1793 | 1992 | 1342 | 1931 | 1896 | 1684 |
| $Z^L_{min}$ | 1306 | 997 | 873 | 1637 | 1414 | 952 | 1793 | 1992 | 1277 | 1874 | 1896 | 1648 |
| $Z^L_{avg}$ | 1361.3 | 997 | 873 | 1640.4 | 1414 | 952 | 1793 | 1992 | 1320.8 | 1909.2 | 1896 | 1671.2 |
| $Z^L_{std}$ | 0.0270 | 0 | 0 | 0.0022 | 0 | 0 | 0 | 0 | 0.0292 | 0.0187 | 0 | 0.0140 |
| $Z^L_{exact}$ | 1374 | 990.5 | 873 | 1642 | 1406 | 952 | 1793 | 1992 | 1303 | 1931 | 1896 | 1661 |
| $Gap_2$ | 0.93% | -0.66% | 0 | 0.10% | -0.57% | 0 | 0 | 0 | -1.37% | 1.13% | 0 | -0.61% |

believe that for medium scale problem, (GFA) is a good heuristic method to solve the follower's problem.

For the second type of tests, we generate 12 smaller-scale random problems as described above to evaluate the gap between the solution obtained by the algorithm (IRGA) and the exact solution. Each random problem has the number of customers is $M$ = 50 and the number of potential facility locations is $N$ = 20, the exact solution of the problem is solved by the enumeration method. The algorithm (IRGA) run 10 times for each problem. The maximum ($Z^L_{max}$), the minimum ($Z^L_{min}$), the average ($Z^L_{avg}$), and the standard deviation ($Z^L_{std}$) of these 10 objective function values are list in Table 3. The $Gap_2$ in Table 3 is defined as follows:

$$Gap_2 = \frac{Z^L_{exact} - Z^L_{avg}}{Z^L_{exact}}.$$ From Table 3, we know that the optimal performance of (IRGA) is good.

The percentage of gap between the solution obtained by (IRGA) and the optimal solution is quite small in all test problems.

For the third type of tests, we generate 9 large-scale examples to evaluate the stability of the algorithm (IRGA). For each example, the algorithm (IRGA) run 10 times to get solutions. All of the results are presented in Table 4. The $Gap_3$ in Table 4 is defined as follows:

$$Gap_3 := \frac{Z^L_{max} - Z^L_{avg}}{Z^L_{max}}.$$ From Table 4, it is obvious that the algorithm (IRGA) is stability both in

the standard deviation of $Z^L$ and the gap between the average and the maximal of $Z^L$.

## 4.2 A quasi-real example

In this subsection, we consider the 49-nodes data set that described in Daskin [34]. In this data set, the 49-nodes denote the capitals of the continental United States plus Washington, DC. The demands are proportional to the 1990 state populations. The latitudes and longitudes of these 49 nodes are given in the data set. So we take the geodesic distances as the $d_{ij}$. Since the

**Table 4. Stability of the algorithm (IRGA) for large scale examples.**

| $(M, N)$ | (100,50) | | | (150,80) | | | (200,100) | | |
|---|---|---|---|---|---|---|---|---|---|
| $(p, q)$ | (5,10) | (10,5) | (10,10) | (10,15) | (15,10) | (15,15) | (15,20) | (20,15) | (20,20) |
| $Z^L_{max}$ | 1116 | 3741 | 2814.5 | 2194.5 | 4861.5 | 3684 | 3612 | 6175 | 4756 |
| $Z^L_{min}$ | 1078 | 3640 | 2665 | 2066 | 4590 | 3527 | 3335 | 5903 | 4491 |
| $Z^L_{avg}$ | 1097.9 | 3672.5 | 2740.6 | 2118.1 | 4724.2 | 3591.9 | 3467.9 | 6017.1 | 4639.9 |
| $Z^L_{std}$ | 0.0131 | 0.0377 | 0.0437 | 0.0382 | 0.0781 | 0.0573 | 0.0902 | 0.0838 | 0.0866 |
| $t_{avg}$ | 22.93 | 9.28 | 20.88 | 130.80 | 45.90 | 98.42 | 285.34 | 154.94 | 209.31 |
| $Gap_3$ | 1.62% | 1.83% | 2.63% | 3.48% | 2.82% | 2.5% | 3.99% | 2.56% | 2.44% |

**Table 5. Optimal solutions for different $(p, q)$ ($\delta = 0.001$).**

| $(p, q)$ | Leader Location | $Z^L$ | Follower Location | $Z^F$ |
|---|---|---|---|---|
| (3,3) | Florida, Illinois, Pennsylvania | 1210 | Georgia, Louisiana, New York | 1260.5 |
| (3,4) | Florida, Illinois, Pennsylvania | 777.9 | Arkansas, Georgia, North Carolina, Washington DC | 1692.6 |
| (3,5) | California, New York, Texas | 647.4 | Connecticut, Idaho, Nevada, Oklahoma, Pennsylvania | 1823.1 |
| (4,3) | Alabama, Georgia, Missouri, New York | 1452.1 | Kansas, Mississippi, Washington DC | 1018.4 |
| (4,4) | Florida, Illinois, Michigan, New York | 1142 | Idaho, Iowa, Tennessee, Washington DC | 1328.5 |
| (4,5) | California, Illinois, New York, Texas | 913.1 | Connecticut, Louisiana, Oklahoma, Pennsylvania, Wisconsin | 1557.4 |
| (5,3) | Alabama, California, Kentucky, Missouri, New York | 1674.2 | Florida, Kansas, Virginia | 796.3 |
| (5,4) | Florida, Illinois, Indiana, New York, Pennsylvania | 1385.8 | Idaho, Kentucky, Missouri, South Dakota | 1084.7 |
| (5,5) | Alabama, Florida, Illinois, New York, Washing DC | 1142.5 | Georgia, Idaho, Iowa, Michigan, Pennsylvania | 1328 |

geodesic distances are relatively large, we set the value of $\alpha$ to 1 to more clearly reflect the effect of the threshold $\delta$. The leader firm aims at opening $p$ facilities and knows that the follower will open $q$ facilities after her action. Each customer's location can be used as a potential facility location both for the leader firm and the follower firm.

For different $p$ and $q$, suppose that the customers following the partially proportional rule with a threshold $\delta = 0.001$, the optimal solutions are presented in Table 5 as follows. From Table 5, the number of the opening facility is most important for the market share captured by the leader firm or follower firm. If $p > q$, then the market share captured by the leader firm large than the market share captured by the follower firm, and vice versa. For a fixed number of $q$, the market share captured by the leader firm increases with the increases in the number of $p$. Conversely, for a given $p$, the market share gained by the leader firm decreases as $q$ increases.

In order to investigate the effect of the threshold $\delta$, let $\delta$ change from 0.001 to 0.010 in steps of 0.001. Suppose that the leader firm has the same number of facilities as the follower firm, that is $p = q = 3$. The optimal solutions corresponding to these different thresholds are listed in Table 6. From Table 6, we know that even if the location of the leader's facilities remains the same, the location of the follower's facilities may change with the threshold $\delta$. From the overall trend, the market share captured by the leader firm $Z^L$ increases with the increases of the threshold $\delta$. Another interesting result is: although the leader firm can choose the facility locations before the follower firm, it does not have a first-mover advantage when the threshold $\delta$ is small.

**Table 6. Effect of the threshold $\delta$ ($p = q = 3$).**

| $\delta$ | Leader Location | $Z^L$ | Follower Location | $Z^F$ |
|---|---|---|---|---|
| 0.001 | Florida, Illinois, Pennsylvania | 1210 | Georgia, Louisiana, New York | 1260.5 |
| 0.002 | Florida, Kentucky, Pennsylvania | 1213.8 | Connecticut, Mississippi, Oregon | 1256.7 |
| 0.003 | Florida, Michigan, Pennsylvania | 1219.2 | Mississippi, New York, Ohio | 1251.3 |
| 0.004 | Florida, Michigan, Pennsylvania | 1214.9 | Alabama, Louisiana, New Jersey | 1255.6 |
| 0.005 | Florida, New York, Washington DC | 1229.5 | Alabama, Louisiana, Pennsylvania | 1241 |
| 0.006 | Florida, New York, Pennsylvania | 1232.8 | Louisiana, Nebraska, Tennessee | 1237.7 |
| 0.007 | California, Florida, Pennsylvania | 1268 | Alabama, Georgia, Louisiana | 1202.5 |
| 0.008 | California, Florida, Pennsylvania | 1280.9 | Georgia, Indiana, Mississippi | 1189.6 |
| 0.009 | California, Florida, Pennsylvania | 1282.4 | Illinois, New York, Oklahoma | 1188.1 |
| 0.010 | California, Florida, Pennsylvania | 1278.9 | Mississippi, Oklahoma, Tennessee | 1191.6 |

**Table 7. Customers completely captured by the leader and the follower firms for different $\delta$ ($p = q = 3$).**

| $\delta$ | Customers captured by the leader firm | Customers captured by the follower firm |
|---|---|---|
| 0.003 | Alabama, <u>Florida</u>, Georgia, <u>Michigan</u>, Indiana, Kentucky, <u>Pennsylvania</u>, Virginia | Connecticut, Illinois, Louisiana, <u>Mississippi</u>, Missouri, New Jersey, <u>New York</u>, <u>Ohio</u>, Vermont, Wisconsin |
| 0.004 | <u>Florida</u>, Georgia, Kentucky, <u>Michigan</u>, Ohio, <u>Pennsylvania</u>, Virginia | <u>Alabama</u>, Arkansas, Illinois, Indiana, <u>Louisiana</u>, Mississippi, Missouri, <u>New Jersey</u>, New York, Tennessee, Wisconsin |

**Table 8. Effect of the number of facilities $p$ for small and large threshold ($p = q$).**

| $\delta$ | Obj | $p = 1$ | $p = 2$ | $p = 3$ | $p = 4$ | $p = 5$ | $p = 6$ | $p = 7$ | $p = 8$ | $p = 9$ | $p = 10$ |
|---|---|---|---|---|---|---|---|---|---|---|---|
| 0 | $Z^L(p)$ | 1277.9 | 999.3 | 873.7 | 1002.6 | 1084 | 1206.3 | 1341.2 | 1384.3 | 1398.8 | 1401 |
| | $Z^F(p)$ | 1192.6 | 1471.2 | 1596.8 | 1468 | 1386.5 | 1264.2 | 1129.3 | 1086.2 | 1071.7 | 1069.5 |
| 0.02 | $Z^L(p)$ | 1287 | 1304.5 | 1317.9 | 1361.3 | 1397.4 | 1456.3 | 1473.8 | 1492 | 1522 | 1558.9 |
| | $Z^F(p)$ | 1183.5 | 1166 | 1152.6 | 1109.2 | 1073.1 | 1014.2 | 996.7 | 978.5 | 948.5 | 911.6 |

The patronizing patterns of the customers between the leader firm and the follower firm when $\delta = 0.003$ and $\delta = 0.004$ are shown in Table 7. In Table 7, the second and the third columns denote the customers completely captured by the leader firm and by the follower firm, respectively. The states with underlines denote the facility locations of the two firms. For each $\delta$, the demands of all remaining customers not mentioned in Table 7 are evenly distributed between the two firms. From Table 7, we find that as $\delta$ changes from 0.003 to 0.004, the leader firm's facility location scheme remains the same, but the follower firm's location scheme is completely changed. If the follower firm's location scheme remains unchanged, i.e. in states of Mississippi, New York and Ohio, then the market share captured by the follower firm is 1187.1. However, by changing its location scheme to new locations of Alabama, Louisiana, and New Jersey. The market share captured by the follower firm can be added to 1255.6.

Suppose that $p = q$, for threshold $\delta = 0$ and $\delta = 0.02$, the effects of different $p$ on the market share captured by the leader firm and the follower firm are presented in Table 8.

In order to illustrate the relationship between the market share and $p$ clearly, the results of Table 8 are shown in Fig 2. From Fig 2, we find that for small $\delta$, the value of $Z^L(p)$ decreases first and then increases with the increases of $p$. For large $\delta$, the value of $Z^L(p)$ increases with the increases of $p$. Suppose that $p = q$, then we can get an interesting conclusion. That is, only for the competitive facility location problem with a large $\delta$ or a large $p$, there is a first-mover advantage for the leader firm.

## 5 Conclusion

In this paper, we studied the competitive facility location problem in the framework of the leader-follower game. The partially proportional rule with a threshold is presented in this paper. This rule is suitable to describe customer behaviors when they are facing the choice of competitive chain facilities that offer exclusive services. A greedy based algorithm is proposed to solve the follower's problem after given the facility locations of the leader. By embedding the greedy-based follower algorithm, an improved ranking based facility location algorithm is given to solve the problem. Numerical tests about the greedy-based follower algorithm and the ranking-based greedy algorithm show that the algorithm proposed in this paper can solve the competitive facility location problem effectively. Through detailed analysis of a quasi-real example, the effects of different parameters on the market share captured by the leader firm

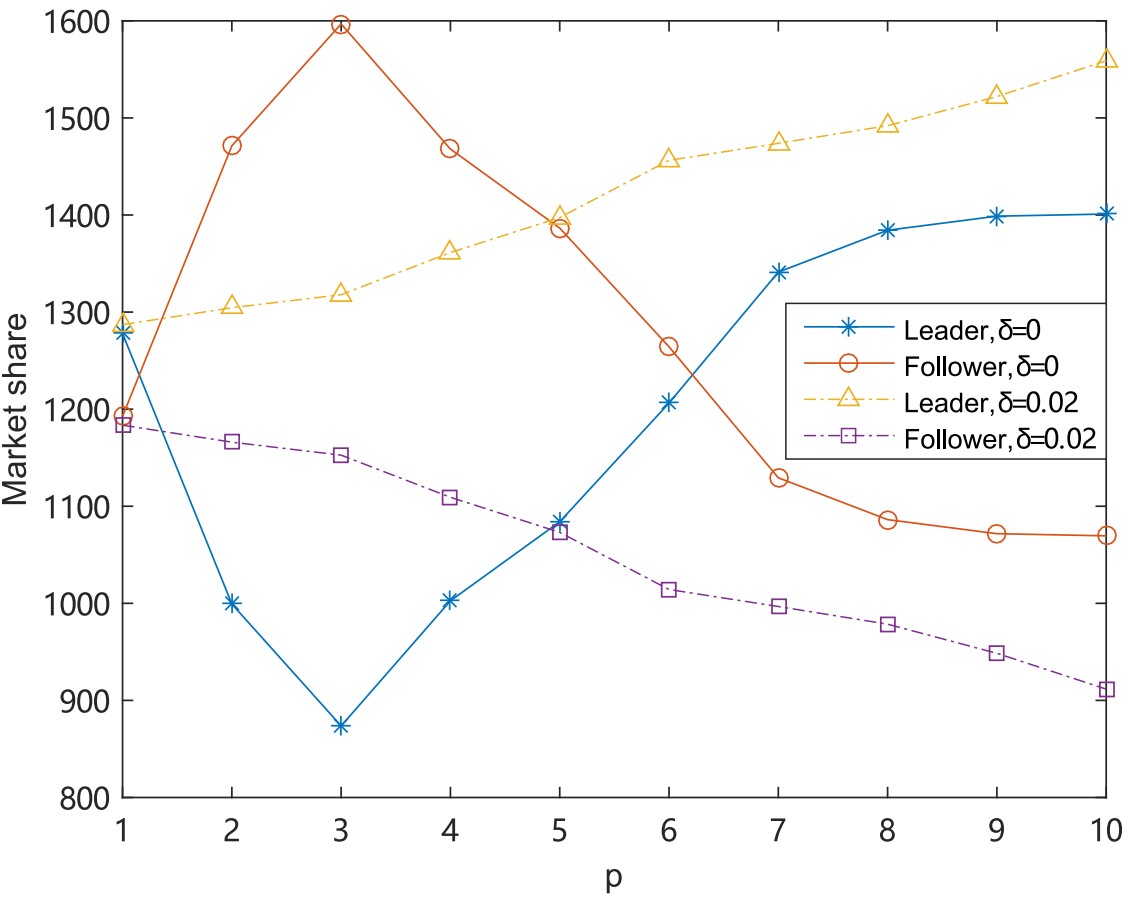

**Fig 2. Example for the partially proportional rule with a threshold.**

and the follower firm are presented. An interesting conclusion is that in some cases, there is no first-mover advantage for the leader firm.

As future directions, one can consider the competitive facility location problem with a limited budget under the partially proportional rule with a threshold. And the uncertainty of the threshold can be considered in the problem. From the computational point of view, the design of effective algorithms to solve these problems is another field for research.

## Supporting information

**S1 File. The data set of the 49-nodes example in subsection 4.2 is provided as S1 File.** The first two columns of the data are latitudes and longitudes of the capitals of the continental United States plus Washington, DC. The third column is the demand $w_i$ and the last column is the name of the states.
(XLSX)

## Acknowledgments

This research was funded by The National Social Science Fund of China (17BGL182).

## Author Contributions

**Conceptualization:** Wuyang Yu.

**Data curation:** Wuyang Yu.

**Formal analysis:** Wuyang Yu.

**Funding acquisition:** Wuyang Yu.

**Investigation:** Wuyang Yu.

**Methodology:** Wuyang Yu.

**Project administration:** Wuyang Yu.

**Resources:** Wuyang Yu.

**Software:** Wuyang Yu.

**Supervision:** Wuyang Yu.

**Validation:** Wuyang Yu.

**Visualization:** Wuyang Yu.

**Writing – original draft:** Wuyang Yu.

**Writing – review & editing:** Wuyang Yu.

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
