## [Decision Letter · Decision Letter 0]

15 Oct 2019

PONE-D-19-21394

A leader-follower model for discrete competitive facility location problem under the partially proportional rule with a threshold

PLOS ONE

Dear Dr. Yu,

Thank you for submitting your manuscript to PLOS ONE. After careful consideration, we feel that it has merit but does not fully meet PLOS ONE’s publication criteria as it currently stands. Therefore, we invite you to submit a revised version of the manuscript that addresses the points raised during the review process.

We recommend that it should be revised taking into account the changes requested by the reviewers. Since the requested changes includes Major Revision, the revised manuscript will undergo the next round of review by the same reviewers.

We would appreciate receiving your revised manuscript by Nov 29 2019 11:59PM. To enhance the reproducibility of your results, we recommend that if applicable you deposit your laboratory protocols in protocols.io, where a protocol can be assigned its own identifier (DOI) such that it can be cited independently in the future. For instructions see: http://journals.plos.org/plosone/s/submission-guidelines#loc-laboratory-protocols

We look forward to receiving your revised manuscript.

Kind regards,

Baogui Xin, Ph.D.

Academic Editor

PLOS ONE

Journal Requirements:

2.  We note that Figures 2 and 3 in your submission contain map images which may be copyrighted. All PLOS content is published under the Creative Commons Attribution License (CC BY 4.0), which means that the manuscript, images, and Supporting Information files will be freely available online, and any third party is permitted to access, download, copy, distribute, and use these materials in any way, even commercially, with proper attribution. For these reasons, we cannot publish previously copyrighted maps or satellite images created using proprietary data, such as Google software (Google Maps, Street View, and Earth). For more information, see our copyright guidelines: http://journals.plos.org/plosone/s/licenses-and-copyright.

a)    You may seek permission from the original copyright holder of Figure(s) [#] to publish the content specifically under the CC BY 4.0 license.

Reviewers' comments:

Reviewer's Responses to Questions

**Comments to the Author**

1. Is the manuscript technically sound, and do the data support the conclusions?

Reviewer #1: Yes

Reviewer #2: Yes

2. Has the statistical analysis been performed appropriately and rigorously? 

Reviewer #1: N/A

Reviewer #2: Yes

3. Have the authors made all data underlying the findings in their manuscript fully available?

Reviewer #1: Yes

Reviewer #2: Yes

4. Is the manuscript presented in an intelligible fashion and written in standard English?

Reviewer #1: Yes

Reviewer #2: Yes

5. Review Comments to the Author

Reviewer #1: - The text is well written and easy to follow and understand.

- The manuscript investigates a new customer behavior model and proposes a new revised heuristic algorithm to solve the problem.

- The author, generally speaking, uses proper English. However, there are some very basic mistakes of the use of English in the text. So, there is a need of proofreading, preferably by a native English speaker.

- The novelty of the paper is the definition of the new customer behavior rule: partially proportional rule with the threshold. More justification is need to convince readers why that type of customer behavior is important to investigate. There may be other papers investigating the same or similar type of behavior in another research area or business case in which that kind of behaviour is applied etc.

- In defining the proposed cust. behavior model, it is mentioned that if the difference in attraction is less than the threshold the customer should choose one of the facilities in proportion to the attraction of the facility. That means if the threshold is 2 and attraction(A) is 10 and Attraction(B) is 9 then the 9 out of 19 of the demand is for B. However, in linear model it is assumed that the demand is equally split. That needs to be corrected.

- Some variables used in heuristic algorithms have never been defined (L, Y(j.N), J(F,N) etc.). That makes it really difficult to follow the model. I could not completely comprehend the IRGA model for that reason.

- In table 2, Gap increases by the increase in q. That may be an important problem when q is even higher. Do you have any explanation for that?

- Alpha value of "2" is used in numerical examples, however, value of "1" is used in quasi-real examples. Why?

Reviewer #2: Please see the attached pdf.

Please see the attached pdf.

Please see the attached pdf.

Please see the attached pdf.

6. PLOS authors have the option to publish the peer review history of their article (what does this mean?). If published, this will include your full peer review and any attached files.

Reviewer #1: No

Reviewer #2: No

---

## [Author Response · Author response to Decision Letter 0]

24 Oct 2019

Reviewer #1: - The text is well written and easy to follow and understand.

- The manuscript investigates a new customer behavior model and proposes a new revised heuristic algorithm to solve the problem.

- The author, generally speaking, uses proper English. However, there are some very basic mistakes of the use of English in the text. So, there is a need of proofreading, preferably by a native English speaker.

Response: We checked the manuscript very carefully and corrected all the mistakes that we found.

- The novelty of the paper is the definition of the new customer behavior rule: partially proportional rule with the threshold. More justification is need to convince readers why that type of customer behavior is important to investigate. There may be other papers investigating the same or similar type of behavior in another research area or business case in which that kind of behaviour is applied etc.

Response: In the revised manuscript, we added a new scene in which customers usually follow the partially proportional rule with a threshold. This helps to convince the reader of the importance of studying this rule.

- In defining the proposed cust. behavior model, it is mentioned that if the difference in attraction is less than the threshold the customer should choose one of the facilities in proportion to the attraction of the facility. That means if the threshold is 2 and attraction(A) is 10 and Attraction(B) is 9 then the 9 out of 19 of the demand is for B. However, in linear model it is assumed that the demand is equally split. That needs to be corrected.

Response: In our definition, two firms with attractiveness gaps less than the threshold are indistinguishable for the customer. So the customer selects any one of the two firms with the same probability. That is to say, from the perspective of the firm, when the attractiveness gap between the leader firm and the follower firm is less than the threshold, the two firms will split the customer's demand equally. For both firms, it doesn't important how the captured demand is allocated in their internal facilities.

- Some variables used in heuristic algorithms have never been defined (L, Y(j.N), J(F,N) etc.). That makes it really difficult to follow the model. I could not completely comprehend the IRGA model for that reason.

Response: We added the definitions of these variables in the revised manuscript and corrected some inaccurate expressions.

- In table 2, Gap increases by the increase in q. That may be an important problem when q is even higher. Do you have any explanation for that?

Response: It can also be seen from Table 2 that for a given solution of the leader firm, the average gap between the sub-algorithm (GFA) and the exact method increases with $q$. Alekseeva et al. [37] observed that under fixed value of $M$ and $N$, the leader-follower competitive facility location problem becomes more difficult when $p=q=[N/3]$. In this case, a lot of calculation time is required to solve the follower's problem and check the feasibility of the bi-level structure. Most of the literature mentioned in Table 1 that considers foresight can only solve small scale problems within $q<10$. So from a perspective of computation time, we believe that for medium scale problem, (GFA) is a good heuristic method to solve the follower's problem. We added the explanation for this point in the revised manuscript.

- Alpha value of "2" is used in numerical examples, however, value of "1" is used in quasi-real examples. Why?

Response: The distances used in numerical texts are relatively small due to the limitation of the region, and the geodesic distances used in the quasi-real example are large, so we set different values of Alpha to more clearly refect the effect of the threshold Delta. We added the explanation for different values of Alpha in the revised manuscript.

Another Note: Since we are unable to obtain the copyright of the maps in Fig.2 and Fig.3, we changed the information of these two figures into Table 7 in the revised manuscript.

Reviewer #2:

This paper is about a new method for modeling the behavior of customers when they cope with a competing market for an exclusive service. To the best of my knowledge, the partially proportional rule with a threshold is not considered in the literature before. I think that the proposed practical idea is novel and therefore, the manuscript deserves publishing.

Major comments

 First paragraph of Section 2: The attraction of a firm should be defined as the sum of the attraction of its facilities; otherwise, the reader would be confused.

Response: We added the definition of the attraction of a firm in the revised manuscript.

 The numbers under the partially proportional rule with threshold δ=1 should be modified. They are not described, but I think they are probabilities of selection. However, they are misleading, since the reader thinks that it is possible that a selection is made between a yellow and a blue facility. This is not possible according to the text. I think you should separate the probability of selection of the firm and the probability of selecting a facility, and the multiplication is misleading.

Response: We separated the probability of selecting a firm from the probability of selecting a facility and modified the representation in Figure 1.

 What do the brackets mean in Equation (1)? If you do not mean the integral part, please use parentheses.

Response: We modified the brackets to be parentheses.

Minor comments

 The sentence “if the total attraction of two firms are equal or approximated”, does not have correct meaning that you want.

Response: We modified all inaccurate expressions that we found. 

 The grammatical errors in the manuscript should be corrected. Some of them are as follows.

 Competition type usually be divided into two categories

 will not take actions against new enter firm

 Section 4 illustrates the effectiveness of the algorithm through numeral tests (it should be numerical tests)

 to the exclusive of these services (exclusiveness)

Response: We checked the manuscript very carefully and corrected all the grammatical errors that we found. 

Another Note: Since we are unable to obtain the copyright of the maps in Fig.2 and Fig.3, we changed the information of these two figures into Table 7 in the revised manuscript.

Journal Requirements:

Response: We modified the original by using the template of PLOS ONE.

2. We note that Figures 2 and 3 in your submission contain map images which may be copyrighted. All PLOS content is published under the Creative Commons Attribution License (CC BY 4.0), which means that the manuscript, images, and Supporting Information files will be freely available online, and any third party is permitted to access, download, copy, distribute, and use these materials in any way, even commercially, with proper attribution. For these reasons, we cannot publish previously copyrighted maps or satellite images created using proprietary data, such as Google software (Google Maps, Street View, and Earth). For more information, see our copyright guidelines: http://journals.plos.org/plosone/s/licenses-and-copyright.

We require you to either (1) present written permission from the copyright holder to publish these figures specifically under the CC BY 4.0 license, or (2) remove the figures from your submission.

Response: Since we are unable to obtain the copyright of the maps in Fig.2 and Fig.3, we changed the information of these two figures into Table 7 in the revised manuscript.

---

## [Decision Letter · Decision Letter 1]

12 Nov 2019

A leader-follower model for discrete competitive facility location problem under the partially proportional rule with a threshold

PONE-D-19-21394R1

Dear Dr. Yu,

We are pleased to inform you that your manuscript has been judged scientifically suitable for publication and will be formally accepted for publication once it complies with all outstanding technical requirements.

With kind regards,

Baogui Xin, Ph.D.

Academic Editor

PLOS ONE

Additional Editor Comments (optional):

Reviewers' comments:

Reviewer's Responses to Questions

**Comments to the Author**

1. If the authors have adequately addressed your comments raised in a previous round of review and you feel that this manuscript is now acceptable for publication, you may indicate that here to bypass the “Comments to the Author” section, enter your conflict of interest statement in the “Confidential to Editor” section, and submit your "Accept" recommendation.

Reviewer #1: All comments have been addressed

Reviewer #2: All comments have been addressed

2. Is the manuscript technically sound, and do the data support the conclusions?

Reviewer #1: Yes

Reviewer #2: Yes

3. Has the statistical analysis been performed appropriately and rigorously? 

Reviewer #1: N/A

Reviewer #2: Yes

4. Have the authors made all data underlying the findings in their manuscript fully available?

Reviewer #1: Yes

Reviewer #2: No

5. Is the manuscript presented in an intelligible fashion and written in standard English?

Reviewer #1: Yes

Reviewer #2: Yes

6. Review Comments to the Author

Reviewer #1: Thank you for addressing the comments and concerns I have raised in my review. I believe the paper deserves publishing.

Reviewer #2: The author has carefuly applied the comments.

I think the manuscript has been significantly improved and can be published in its current form.

7. PLOS authors have the option to publish the peer review history of their article (what does this mean?). If published, this will include your full peer review and any attached files.

Reviewer #1: No

Reviewer #2: Yes: Mohammad Mahdi Nasiri

---

## [Editor Report · Acceptance letter]

18 Nov 2019

PONE-D-19-21394R1 

A leader-follower model for discrete competitive facility location problem under the partially proportional rule with a threshold 

Dear Dr. Yu:

I am pleased to inform you that your manuscript has been deemed suitable for publication in PLOS ONE. Congratulations! Your manuscript is now with our production department. 

With kind regards,

on behalf of

Prof. Baogui Xin 

Academic Editor

PLOS ONE